# Reshaping cancer cell plasticity by P-body dynamics and protein translation

Emanuela Senatore, Laura Rinaldi, Francesco Chiuso, Antonio Giuseppe Bianco & Antonio Feliciello ✉

Processing bodies (P-bodies) are membrane-less organelles composed of condensed mRNAs and proteins that play essential role in mRNAs decay and storage, contributing to the translational control of cellular proteostasis. Regulation of P-body assembly/disassembly by signaling events, cellular stress or specific environmental conditions shapes the rate of RNA turnover and protein synthesis, controlling cell growth, differentiation and survival. Deregulation of protein translation is an important factor for tumor development and progression and cancer cells benefit from P-bodies to reshape their proteome to support specific metabolic needs and promote tumor development, progression and metastasis. Hence, understanding the composition and the regulation of P-bodies, both under physiological and pathological conditions, will define the mechanisms underlying cancer cell plasticity and develop novel therapeutic strategies to inhibit cancer growth and metastasis. Here, we will discuss the principal mechanisms of P-body regulation and function, with special focus on the role of these ribonucleoprotein condensates in cancer.

The regulation of gene expression is an important mechanism that cells adopt to modulate their proteome composition under specific growth and environmental conditions, thus playing a fundamental role in cell proliferation, differentiation, metabolism, and survival. Gene expression is controlled at multiple levels: epigenetic, transcriptional, and post-transcriptional, each involving spatially and timely regulated distinct molecular mechanisms. The first level of regulation involves epigenetic modifications of chromatin that include methylation of DNA and post-translational modifications (PTM) of histones that deeply influence chromatin structure and gene expression[1]. The second level is mediated by transcription factors that bind to DNA sequences located within promoter genes and to components of the transcription machinery and regulate the rate of gene transcription[2,3]. The third level of regulation takes place after mRNA synthesis and includes processing of newly synthetized transcripts, their export from the nucleus to the cytoplasm, association with ribosomes, and degradation by ribonucleases. These distinct regulatory layers, even if spatially and mechanistically different, are essential to define the final cellular proteome, and their deregulation is causally linked to the development of a variety of human disorders, including cancer[4]. Recently, a regulatory role of gene expression has been attributed to the mechanism of liquid–liquid phase sepration (LLPS), a physical process of condensation of macromolecular complexes at distinct intracellular sites[5]. At the epigenetic level, the formation of chromatin-associated condensate allows the organization of regulatory factors in the nuclear space[6]. The transcriptional control is often modulated by nuclear transcriptional condesates, that concentrate transcription machinery components at active gene loci[7]. At the post-

transcriptional level, the formation of cytoplasmatic ribonucleoprotein (RNP) condensates, including mRNAs and proteins, namely P-bodies, allows a fine regulation of mRNA transcripts fate[8] (Fig. 1). At P-bodies, mRNAs can be temporarily stored or processed by the RNA decay machinery. This is a complex regulatory system of RNA translational repression and turnover that dynamically shapes the cellular proteome[9]. Current evidence indicates that the dynamic assembly/disassembly of P-bodies constitutes an efficient mechanism that cells adopt to control the rate of protein translation during cell cycle progression or under specific cellular needs and environmental conditions[10–14]. Given the importance of P-bodies in the control of RNA metabolism, decay, and translation, these condensates are attracting great interest not only in the field of cell biology but also in translational oncology. This minireview summarizes the most relevant and recent findings on the role of P-bodies in cancer biology, highlighting the possibility of targeting the RNPs condensates for the treatment of aggressive malignant tumors.

## Membrane-less organelles and liquid–liquid phase separation (LLPS)

An important control mechanism of protein translation is represented by the binding of mRNA molecules to RNA binding-proteins (RBPs), leading to the formation of RNP complexes[15]. Under specific cellular conditions, different RNPs can physically condense to form a variety of membrane-less macromolecular complexes, including processing bodies (P-bodies) and stress granules[16]. These condensates form through the LLPS process driven by weak interactions between proteins and nucleic

Department of Molecular Medicine and Medical Biotechnology, University Federico II, Naples, Italy. ✉e-mail: feliciel@unina.it

**Fig. 1 | LLPS-mediated regulation of gene expression.** The first level of regulation of gene expression is represented by modifications and remodeling of chromatin structure that occur within the nucleus. This mechanism is tightly controlled by chromatin-associated condensates (blue) that assemble regulatory factors including DNA methyltransferases (DNMTs), histone deacetylase (HDAC), histone acetyltransferase (HAT), and heterochromatin protein 1 (HP1). The second level of regulation concerns the transcription of unmethylated genes by the RNA polymerase machinery. Transcription factors (TFs), transcriptional co-activators (TF-A), and repressors (TF-R), included in the nuclear condensates (grey), contribute to the regulation of gene transcription. The third level of regulation of gene expression is represented by the control of mRNA stability and translation that occurs within the cytoplasm. Here, mature mRNA molecules exported from the nucleus accumulate within membrane-less condensates, namely P-bodies (green) and stress granules (yellow). In these compartments, ribonucleoprotein complexes (RBPs) tightly control the translation and turnover of mRNAs in response to specific cellular needs.

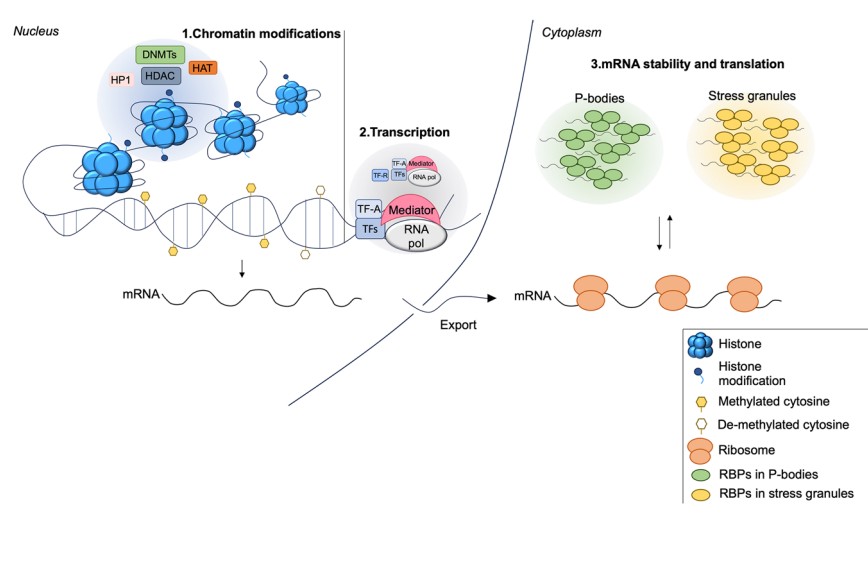

acids, playing a fundamental role in localizing and concentrating untranslated mRNAs at specific intracellular compartments[17–23]. Besides P-bodies and stress granules, other mRNPs granules have been identified in eukaryotic cells, including the *TIS granules* composed of mRNA with an AU-rich element associated with the endoplasmic reticulum[24], the *Cajal bodies* that are subnuclear structures linked with the nucleolus[25] and the U snRNP (Uridine-rich small nuclear RNP) bodies, also named *U-bodies*, that associate with P-bodies and are sites of assembly of snRNPs in the cytoplasm before their transport to the nucleus[26,27]. In *Drosophila*, alterations of P-body structure impact on the composition of *U-bodies*, suggesting a cooperation between both types of organelles in the regulation of snRNPs[26]. Moreover, *GW bodies* are cytoplasmatic foci that regulate the storage, the processing, and the stability of mRNAs, and their dynamic is closely connected to cell cycle phase[28,29]. GW bodies were originally identified as granules containing the GW182 protein involved in miRNA-mediated gene silencing and were considered similar to P-bodies[30]. Subsequent studies in *Drosophila* revealed that GW bodies have different dynamics and protein composition compared to P-bodies, suggesting that they are distinct mRNPs granules[31]. However, this conclusion has not been formally demonstrated in human cells.

P-bodies are constitutively present in cells, whereas the assembly of stress granules is primed by stress conditions. Although morphologically and biochemically distinct, P-bodies and stress granules are functionally and dynamically interconnected. In particular, RBPs and RNA molecules located at the surface of these condensates mediate the docking and the consequent shuttling of mRNAs and RBPs between both compartments[21,32]. DDX6, a principal component of P-bodies, plays a fundamental role in this process by limiting stress granule formation and negatively regulating the docking between the two types of mRNP aggregates[33].

## P-body background and historical context
The complexity and composition of P-bodies have been well characterized through studies in lower eukaryotes. The principal constituents of P-bodies are proteins involved in the translational repression and mRNA decay that include deadenylating (CCR4-Not) and decay (Lsm1-7) protein complexes, decapping enzymes (DCP1/DCP2), activators of decapping enzymes (EDC3/EDC4 and PAT1), the RNA helicase (DDX6), and ribonuclease (XRN1)[34–36]. P-bodies also contain microRNAs (miRNAs), mRNA-binding proteins, and regulators involved in translational repression, such as

CPEB3[37,38]. The translational initiation factor eIF4E and the related factor eIF4E-transporter (eIF4E-T), both involved in the translational repression, interact with DDX6 and localize at P-bodies[39]. Interestingly, eIF4E is the only component of the translation machinery localized at P-bodies, where its interaction with 4E-T represses protein translation[40]. At first, P-bodies were considered sites of mRNA turnover, since the principal components of P-bodies in *yeast* were proteins involved in mRNA decay[41]. Subsequent studies demonstrated that P-bodies are also sites for the temporary storage of untranslated transcripts. In fact, reporter mRNAs can be temporarily localized in P-bodies when the translation is inhibited. The removal of protein synthesis inhibitors promotes a rapid exit of mRNAs from the mRNP aggregates and resumption of protein synthesis[42]. Moreover, cells arrested in the stationary phase have larger P-bodies, mostly enriched of mRNAs that become available for translation when cells re-enter the growth phase[42]. In *metazoans*, the translation of the cationic amino acid transporter (CAT-1) transcript, that is repressed by miR-122, is restored after its release from P-bodies and consequent recruitment by polysomes[43].

The majority of early studies on P-bodies were conducted in yeast cells, which exhibit several differences from mammalian P-bodies, most notably the regulation of assembly. In particular, in these cells, P-bodies are not present under normal growth conditions, and their assembly can be primed by different environmental conditions, such as glucose deprivation, osmotic stress, and DNA replication stress[44]. However, these events have not been confirmed in all eukaryotes, where P-bodies are constitutively present. Accordingly, since most of the studies and findings made in yeast cells cannot be extended to mammalian systems, this review will primarily focus on the structure and regulation of mammalian P-bodies.

## Key players of mammalian P-bodies
Although the number of resident and regulatory proteins that localize at P-bodies is still growing, in *mammals*, only a few proteins are essential for their assembly, such as DDX6 and its partners 4-ET and LSM14A[45,46]. Genetic knockdown experiments demonstrated that DDX6, 4-ET, CCR4, and LSM1 are necessary for their mutual accumulation in P-bodies[39]. Other proteins, although playing an essential role in the regulation of mRNAs within P-bodies, are not required for the assembly of these condensates. This is the case of DCP2 and EDC3, whose silencing in mammalian cells does not affect the number of P-bodies[39,45]. An important regulatory element in P-body assembly is the physical interaction between various components of

**Fig. 2 | Mechanisms of P-body dynamics.** P-bodies are intracellular condensates composed of mRNAs and RNA-binding proteins (RBPs), including deadenylating protein complex CC4-Not, decay complex Lsm 1-7, decapping enzymes DCP1/2, activators of decapping enzymes EDC3/4 and PAT1, RNA helicase DDX6, and ribonuclease XRN1.
**A** Under stress conditions, Jnk phosphorylates 4-ET and promotes P-body assembly. Ubiquitylation of DCP1a by TRAF6 and subsequent phosphorylation by Jnk induces its binding to EDC4 and XRN1. Dimethylation of Lsm4 and ubiquitylation of 4-ET by Trim56 promotes P-body assembly.
**B** Ubiquitylation of EDC4 by Trim24 inhibits the assembly of the decapping protein complex. Otud4-mediated deubiquitylation of 4-ET promotes P-body disassembly.

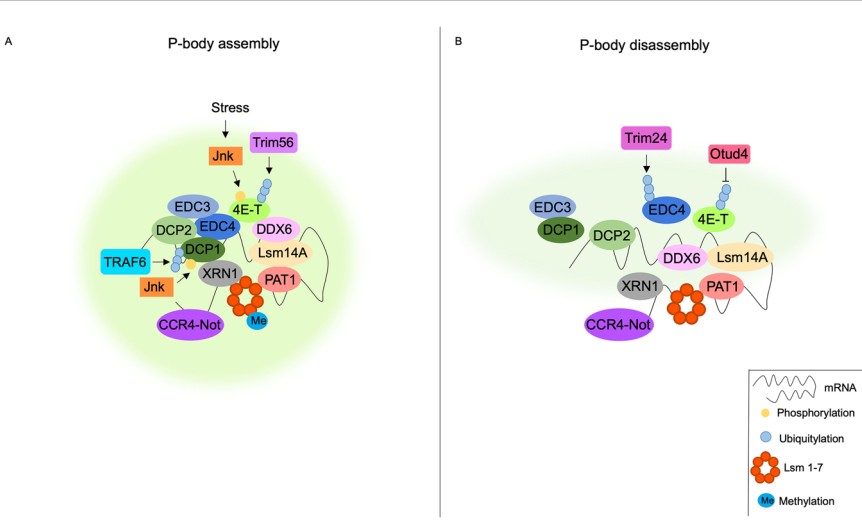

these granules, as demonstrated by a two-hybrid fluorescence-based assay[47]. The N-terminal portion of EDC4 directly interacts with LSM14A, PATL1, XRN1 and the Negative Regulator Of P-Body Association (NBDY), while only indirectly with DDX6, whereas the C-terminal portion of the protein binds to DCP1A and CCHR1 and indirectly interacts with EDC3. In cells lacking P-bodies, as a consequence of genetic deletion of DDX6 or LSM14A, overexpression of the N-terminal portion of EDC4, but not wild-type form, is sufficient to form cytoplasmic dots that are structurally indistinguishable from P-bodies, but lack LSM14A and DDX6, respectively[47]. The interaction between EDC4 and XRN1, an mRNA decay enzyme, is also essential for P-body dynamics and for the regulation of mRNA decapping and decay. Thus, in human cells depleted of XRN1, P-bodies are enlarged. Re-expression of a catalytic mutant of XRN1 that retains the mRNA decay activity in XRN1-depleted cells is not sufficient to restore the normal size of P-bodies. Instead, the effects of XRN1 depletion on P-bodies size can be reversed by overexpressing both the catalytic and EDC4-interacting domains of the protein. This indicates that EDC4-XRN1 interaction is essential for P-body dynamics and mRNA decay activity in human cells[48]. Conversely, the C-terminal portion of EDC4 is required for EDC4 self-association and condensation, and its overexpression is sufficient to promote the assembly of P-bodies[49]. Altogether, these results suggest that the direct interaction among components of RNP complexes is, indeed, a driving force for P-body assembly.

## Regulation of P-body dynamics

Although in *mammals* P-bodies are constitutively present within cells, different conditions can impact on their assembly. In fact, the availability of mRNAs represents an important element for the formation of P-bodies, as demonstrated by the dissolvement of mRNP aggregates treating the cells with actinomycin-D, a potent and selective inhibitor of RNA transcription[50]. Moreover, experiments aimed to reconstitute bioengineered condensates in human cells demonstrated that the recruitment of RNAs influences the composition and size of RNP aggregates, and the effects of RNAs on granule size inversely correlates with its surface density[51].

Metabolic changes can also impact on the assembly of P-bodies. Thus, treatment with 2-deoxiglucose and oligomycin, an experimental condition that reduces intracellular ATP levels and induces energy stress, increases the number of P-bodies[52]. Similarly, treatment with arsenite, an inducer of oxidative stress, promotes P-body assembly[32]. In this context, EDC3 plays a fundamental role. Thus, the mutagenesis of tyrosine 475 on EDC3 alters the binding to DDX6 and DCP1A and markedly increases P-body number in response to arsenite treatment, supporting a critical role of EDC3 in P-body dynamics in response to stress[53]. Cell cycle progression also influences the number and size of RNP condensates. In cycling cells, P-bodies dissolve

during mitosis, assemble in G1, and enlarge at the interface, with no major changes in the number of these condensates[11].

As firstly observed in *yeast*, also in *mammals* has been proved that P-bodies are temporary stores of untranslated mRNAs, and not only sites of RNA turnover. Thus, in human cells, during amino acid starvation, mRNAs accumulate within P-bodies. Amino acid supplementation promotes the clearance of P-bodies and degradation of selected transcripts[54]. Similarly, under glucose deprivation, mRNAs condensate in P-bodies. Here, a group of transcripts is safely stored, whereas others undergo to degradation[55]. In summary, P-bodies represent a dynamic and rapid mechanism to control mRNA translation and turnover, explaining the variability in the RNA/protein composition of P-bodies in different cell types or under specific biological conditions.

## Control of P-body dynamics by post-translational modifications

Post-translational modifications (PTMs) of proteins play a relevant role in the formation of P-bodies. The decapping enzyme DCP1a is ubiquitylated by TRAF6 and subsequently phosphorylated by c-Jun N-terminal kinase (Jnk) on serine 315. Phosphorylation primes DCP1a binding to EDC4 and Xrn1 and assembly of P-bodies[56,57]. Conversely, the decapping protein complex is negatively regulated by Trim24-mediated ubiquitylation of EDC4[58]. Similarly, ubiquitylation of 4-ET by Trim56 or its deubiquitylation by Otud4 promotes P-body assembly or disassembly, respectively[59]. Ubiquitylation is not the only PTM that regulates P-body dynamics. Dimethylation of two arginine residues within the RGG domain of Lsm4 primes P-body assembly[60]. Under stress conditions, phosphorylation of 4-ET by JNK kinase promotes P-body condensation[61].

cAMP-dependent protein kinase (PKA) also plays a relevant role in P-body dynamics. Thus, in *yeast*, activation of cAMP signaling inhibits P-body formation through PKA-mediated phosphorylation of Pat1 and consequent inhibition of DDX6 recruitment to the complex[62]. Conversely, in *mammals*, activation of PKA induces phase-separation of regulatory subunits of PKA (PKARIα) and consequent formation of biomolecular condensates enriched of PKA and cAMP levels, contributing to compartmentalized cAMP signaling[63]. Similarly, the rise of cAMP levels drastically increases the number of P-bodies, confirming the positive role of PKA in the assembly of the cytoplasmic mRNP granules[64] (Fig. 2).

## A role of P-bodies in cancer development and progression

Derangement of translational regulation is one of the main aspects of cancer cell plasticity, a mechanism that allows tumoral cells to rapidly adapt to changes of microenvironmental conditions[65]. Given the contribution of P-bodies in different aspects of RNA metabolism, decay, and translation, they exert a multifaceted role in cancer growth and dissemination. In fact, by

modulating the expression levels of oncogenes, oncosuppressors, and growth regulators, these RNP condensates support cancer cell adaptation to stress or to specific metabolic needs[66]. The importance of P-bodies in tumor growth derives from the observation that, in tumoral tissues, the expression of fundamental components of these mRNP granules is often deregulated, even in the absence of major changes in P-body dynamics. This is the case of EDC4, an important component of P-bodies, that has been found mutated in BRACA1/2 mutation-negative breast cancer tissues[67]. Also, DCP1a expression is increased in colorectal cancer tissues, and its levels correlate with patients' survival[68]. Furthermore, the metastatic lymph node 51 (MLN51) protein, an RNA-binding protein of the exon junction complex that shuttles mRNA molecules between nucleus and cytoplasm, has been recently identified as a component of P-bodies in human cells, where its overexpression promotes the disassembly of these RNP condensates. Interestingly, MLN51 is overexpressed in HER2$^+$ breast cancer cells, and its expression levels correlate with the abundance of P-bodies. In about 50% of HER2+ breast cancers, MLN51 is co-amplified with c-erbB2, a gene encoding for the HER2 receptor. In breast cancers overexpressing both HER2 and MLN51 proteins, the number of P-bodies is notably decreased, compared to HER2$^+$ breast cancer tissues that do not overexpress MLN51 or to normal tissue counterparts. Although a formal proof of a causal link between P-body disassembly and stages of tumorigenesis or tumor aggressiveness is missing, this study strongly suggests a correlation between P-body dynamics and malignant tumor behavior[69].

A link between P-bodies and tumor progression has been proved in myeloid leukemia (AML) cells that possess an increased number of P-bodies that are essential for tumor progression. In AML cells, overexpression of DDX6 induces sequestration and translational repression of mRNAs encoding for tumor suppressors. Dissolution of P-bodies induces a resumption of mRNA translation and consequent inhibition of AML cell proliferation[70]. This finding demonstrated that, at least in part, the role of P-bodies in cancer cells relies on the translational regulation of selected mRNAs. Similarly, in different types of human cancer, the disassembly of P-bodies, induced by treating cells with an NBDY synthetic peptide that prevents EDC4 condensation, activates the p53 pathway and increases the stability of associated transcripts, thus inhibiting tumor proliferation[49]. These findings indicate that the reversible storage of mRNAs within P-bodies, thus, represents an important homeostatic control mechanism of gene expression underlying cell differentiation and tumorigenesis.

## Regulation of P-body dynamics by signaling pathways in cancer

Alterations of signaling pathways and related PTMs are common features of cancer cells[71]. In this context, derangement of signaling events controlling P-body dynamics and protein translation can be causally linked to cancer development. Thus, Pim1/3 kinases phoshorylate EDC3 on serine 161, originally identified as an AKT phosphorylation site[72–74]. Phoshorylation by Pim1/3 kinases inhibits EDC3 localization at P-bodies, markedly impacting on tumor growth[72]. In prostate cancer cells, as well as in multiple breast cancer cell lines, EDC3 protein is hyperphosphorylated compared to normal cells. In a xenograft mouse model of prostate cancer, overexpression of a phosphorylation-deficient mutant of EDC3, carrying a substitution of serine 161 to alanine, reduces the growth, motility, and invasiveness behavior of cancer cells. As expected, treatment with a small kinase inhibitor that prevents EDC3 phosphorylation increases the number of P-bodies and negatively impacts on tumor growth[72]. The TGFβ pathway has also been correlated to the regulation of P-bodies in cancer. TGFβ promotes the epithelial-to-mesenchimal transition (EMT) in different types of human tumors, including breast cancer[75,76]. Treatment of breast cancer cells with TGFβ or overexpression of Twist, a transcription factor and downstream target of the TGFβ pathway, markedly increases the number of P-bodies. These effects are reversed by removing the growth factor from the medium and require the activation of the autophagic machinery. The induction of EMT by TGFβ relies, in part, on P-body formation. Thus, in orthotopic models of breast cancer, downregulation of DDX6 prevents P-body assembly and inhibits the formation of pulmonary metastasis, thus

highlighting a fundamental role of these RNP condensates in the invasiveness potential of breast cancer cells[77].

The Hippo pathway is an oncosuppressive signaling cascade that negatively controls cancer cell growth. In particular, activation of the Hippo pathway promotes phosphorylation and proteolytic degradation of transcriptional regulators YAP and TAZ. In proliferating cells, the inactivation of the Hippo pathway leads to YAP/TAZ translocation to the nucleus and consequent transcription of genes underlying cell growth and survival[78]. YAP/TAZ have a pro-tumorigenic role in different types of tumors, including colorectal cancer (CRC), and are therapeutic targets for cancer treatment[79–81]. The pro-tumorigenic role of YAP and TAZ depends, at least in part, on their capacity to regulate P-body dynamics. Specifically, YAP/TAZ transcriptionally regulate the levels of mRNAs encoding for SMAD4A, AJUBA, and PNRC1 that accumulate within P-bodies. By controlling the levels of these transcripts, YAP/TAZ promote the assembly of P-bodies. In particular, YAP negatively regulates the expression of PNRC1, a tumor suppressor gene that inhibits P-body assembly. Disassembly of P-bodies induced by the downregulation of DDX6 or LSM14A attenuates the oncogenic role of YAP in colon cancer cells, affecting both proliferation and migration, indicating that P-body dynamics is crucial for tumorigenesis[82].

A role of G protein-coupled receptor signaling cascade in P-body dynamics has been recently identified. Thus, activation of the cAMP-PKA pathway induces a non-proteolytic ubiquitylation of DDX6 by praja2, an E3 ligase involved in different biological processes, both under physiological and pathological conditions, including cancer[83–88]. Ubiquitylation of DDX6 by praja2 markedly impacts on P-body formation and protein translation in human glioblastoma (GBM) cells. In xenograft models of GBM, overexpression of a DDX6-defective ubiquitylation point mutant impairs the formation of P-bodies and accelerated tumor growth[64]. This study points to a relevant link between P-body dynamics and GBM growth, highlighting P-bodies as potential therapeutic targets for the treatment of this aggressive brain tumor.

Ubiquitylation of P-bodies resident proteins plays a relevant role also in CRC. As previously mentioned, in CRC cells, energy stress induced by low ATP levels promotes P-body formation and sustains tumor growth. Mechanistically, ATP depletion induces Trim23-mediated non-proteolytic ubiquitylation of HCLS1-associated protein X-1 (HAX1), a P-body component that binds the 3′UTR of mRNAs[89,90]. Ubiquitylation primes liquid–liquid phase separation of HAX1 that, in turn, induces condensation of DDX6 and LSM14A complexes and P-body assembly with consequent inhibition of protein synthesis. Depleting Trim23 inhibits P-body formation and restrains the proliferation of CRC cells[52]. Altogether, these studies point to a pathogenic role of signaling pathways and PTMs operating on P-body dynamics in the control of cancer growth (Fig. 3).

## P-bodies, hypoxia, and metabolic reprogramming in cancer

Hypoxia is one of the main characteristics of solid tumors, and it is associated with a high grade of malignancy. Hypoxia is caused by a limited vascular oxygen supply within the tumoral tissue that is not proportional to the high rate of cell proliferation[91]. Under hypoxic conditions, cancer cells reprogram their metabolism by adopting aerobic glycolysis and lactic acid fermentation to produce ATP and biosynthetic intermediates. To promote the metabolic rewiring during hypoxia, cancer cells also modify their translational landscape[92]. In this context, P-bodies may play a relevant role, representing an important proteostatic mechanism for cancer cells to survive and proliferate under extreme environmental conditions.

Although a link between hypoxia, P-bodies, and cancer has still not well established, different studies suggest an involvement of mRNPs granules in hypoxia-induced translational and metabolic rewiring of tumor cells. The first functional connection with hypoxia derives from the identification of P-bodies as regulators of HIF1α mRNA fate. HIF1α is the master transcriptional regulator activated during hypoxia. The ubiquitin-specific protease 52 (UPS52), also known as poly(A) nuclease 2 (PAN2), is a component of P-bodies that interacts with- and targets HIF1α mRNA within the RNPs condensates. Localization at P-bodies prevents HIF1α mRNA degradation

**Fig. 3 | Role of P-body dynamics in cancer.** Under energy stress conditions, Trim23 ubiquitylates HAX1 that induces the binding between DDX6 and Lsm14A, favoring the assembly of P-bodies and colon rectal cancer growth. TGF-β promotes P-body assembly that positively correlates with breast cancer growth. cAMP induces the praja2-mediated ubiquitylation of DDX6 that stimulates P-body assembly and regulates glioblastoma. Phosphorylation of EDC3 by Pim1/3 kinases inhibits its localization at P-bodies, preventing their assembly and favoring the growth of prostate cancer. P-body disassembly causes protein translation of ADAR1 mRNA that, in turn, supports gastric cancer growth. Treatment with MEK inhibitors (MEKi) induces P-body dissolution and consequent translation of RAS mRNA that supports chemoresistance.

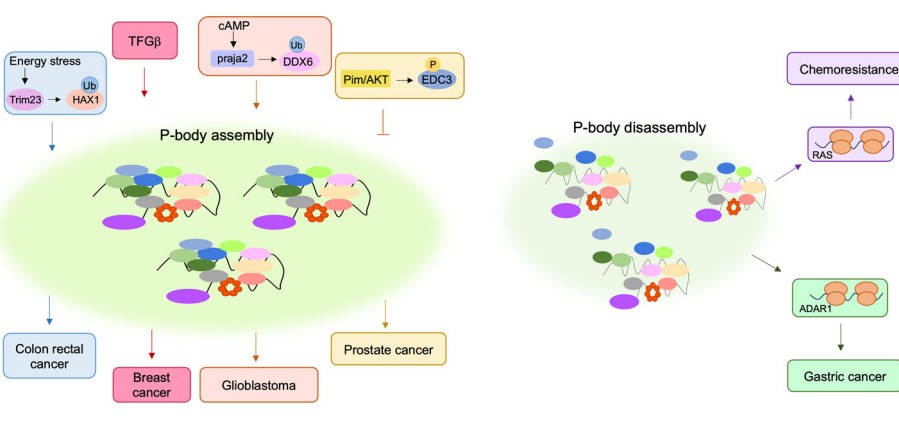

and promotes accumulation of HIF1A protein, leading to transcriptional upregulation of hypoxia-induced target genes. The disassembly of P-bodies by genetic downregulation of *GW182* or *LSM1* reduces the levels of HIF1α mRNA and attenuates the hypoxia-mediated translational reprogramming[93]. On the other hand, hypoxia can negatively regulate protein synthesis through a biphasic inhibition of mRNA translation by two distinct mechanisms. In the acute phase, hypoxia induces phosphorylation and inhibition of the eukaryotic translation initiation factor 2 A (eIF2a). Under prolonged hypoxia, disruption of cap-binding complex eIF4F and sequestration of eIF4E by its inhibitor 4E-BP1 and transporter 4E-T within P-bodies contributes to its reduced translation and regulation of hypoxic gene expression[94]. The biological effects of P-bodies on the translational reprogramming induced by hypoxia have also been demonstrated in cancer cells. In MCF7 breast cancer cells, DDX6 controls the translation of VEGF, a proangiogenic factor that promotes the formation of new vessels during hypoxia. Under normoxic conditions, DDX6 binds to- and localizes VEGF mRNA within DCP1a-containing granules, most likely P-bodies, thus inhibiting its translation. In the course of hypoxia, the decrease of DDX6 levels removes the inhibitory constraint and promotes VEGF translation[95].

A link between hypoxia and P-bodies in the control of the metastatic spreading of cancer cells has been proposed. LIN28A is an RNA-binding protein upregulated in many tumors, and its levels correlate with a poor prognosis. In colon cancer cells, hypoxia increases LIN28A mRNA levels, rather than protein levels, and promotes its localization at P-bodies. Within these condensates, the translation of LIN28A mRNA is inhibited. Interestingly, proteomic analysis revealed that non-coding LIN28A mRNA upregulates the expression of methionyl aminopeptidase 2 (METAP2), a positive regulator of cancer metastasis. Concomitant downregulation of DICER, a ribonuclease involved in the production of miRNAs and siRNAs molecules, abolished the effects of non-coding LIN28A on METAP2 protein expression and cancer metastasis. These findings support a role of P-bodies in the control of a miRNA-mediated mechanism operating in metastatic cancer cells under hypoxia[96]. A relevant role for tumor growth under hypoxia is played by EXOSC9, a component of the RNA exosome complex involved in mRNA decay. Depletion of EXOSC9 in breast cancer cells inhibits proliferation under stress conditions[97]. Although not a resident of P-bodies, EXOSC9 regulates P-body dynamics since its genetic downregulation decreases the number of these condensates. At the mechanistic level, EXOSC9 regulates the decay of APOBEC34 mRNA that is located at P-bodies. Accumulation of this mRNA has a negative effect on P-body assembly, whereas its depletion promotes P-body formation. The assembly of P-bodies regulated by EXOSC9 positively correlates with breast cancer growth. Thus, EXOSC9 depletion attenuates P-body formation and drastically reduces tumor growth in xenograft models of breast cancer. Re-expressing wild-type EXOSC9, but not its mRNA binding mutant, unable to promote P-body condensation, restores cell proliferation and tumor growth[97]. These findings strongly support the role of P-bodies in the

dynamic regulation of protein translation underlying growth and dissemination of cancer cells under stress conditions. Nevertheless, additional studies are needed to provide formal proof of the involvement of P-bodies in the metabolic rewiring of cancer cells under hypoxic or stress conditions and to mechanistically dissect the contribution of P-bodies in the control of global or selective mRNA translation.

## A role of P-bodies in cancer stemness

Cancer stemness is one of the hallmarks of tumor development[98]. Cancer stem cells (CSCs) are responsible for tumor initiation, self-renewal, differentiation, and plasticity[99]. The RNA editing process has been identified as a mechanism of CSCs regulation and, in this context, a relevant role is played by the adenosine deaminase acting on RNA 1 (ADAR1), a member of the double-stranded RNA-specific adenosine deaminase family that converts adenosine to inosine in a double-stranded RNA substrate. ADAR1 exists in two isoforms, both involved in the editing of dsRNA: p100 isoform that is prevalently located in the nucleus, and the p150 principally located in the cytoplasm and inducible by interferon[100]. ADAR1, carrying out its activity in CSCs, plays a pro-tumorigenic role in chronic myeloid leukemia, glioblastoma, and liver cancer[101–103]. An oncogenic activity of ADAR1 has been identified in gastric cancer, where its overexpression promotes tumor growth. Therefore, inhibition of ADAR1 activity has been proposed as a therapeutic strategy for this type of malignant tumor[104,105]. Cytoplasmic ADAR1 protein was originally identified as a resident of stress granules[106]. Recent protein–protein interaction (PPI) network analysis demonstrated that ADAR1 p150 variant forms multimeric complexes with proteins mostly enriched in P-bodies, suggesting a role of this RNA editing enzyme also in P-bodies[107]. The translation of ADAR1 is regulated through the localization of its mRNA within P-bodies. The cytoplasmic polyadenylation element-binding protein 3 (CPEB3) is downregulated in gastric cancer tissue, and its expression correlates with a better prognosis for patients. The oncosuppressive effect of CPEB3 is related to its capacity to promote the localization and translational repression of *ADAR1* mRNA within P-bodies. In mouse models of gastric cancer, overexpression of CPEB3 by adeno-associated vectors inhibits RNA editing by ADAR1 and restrains tumor growth[108] (Fig. 3). Considering the critical role of ADAR1 in different types of CSCs, these findings support an involvement of P-bodies in the regulation of cancer stemness through the control of *ADAR1* mRNA metabolism. In P-bodies, DDX6 binds to- and translationally represses mRNAs encoding for transcription factors that control the exit from the pluripotency state of CSCs[109]. Moreover, a transcriptomic profiling of coding and non-coding RNAs that accumulate in P-bodies shows differentially enriched transcripts at different stage of development in multiple vertebrate species. Importantly, this study revealed the essential role of P-bodies not much in the regulation of the global gene expression, but rather in controlling the accumulation and translational repression of transcripts relative to the previous developmental stage in a miRNA-dependent mechanism. In this context, the cell fate

transition in pluripotent stem cells can be induced by regulating the selective sequestration of RNA in biomolecular condensates at P-bodies[110]. Currently, not enough studies have been conducted to provide formal evidence of the role of P-bodies in cancer stemness. However, the data available so far suggest that P-bodies, through the regulation of the stability and/or translation of specific transcripts, indeed play a crucial role in maintaining the stem-like properties characteristic of cancer cells.

## P-bodies in cancer chemoresistance

Chemoresistance represents one of the most difficult challenges for cancer treatment, demanding the necessity of a deep understanding of the molecular basis of its insurgence that might contribute to develop new drugs that sensitize cancer cells to chemotherapeutic agents. Chemoresistance relies on different molecular mechanisms underlying DNA repair, regulation of oncogenes or oncosuppressors, EMT and cancer stemness[111]. The development of chemoresistance is linked to genetic and epigenetic mechanisms, and it is strictly dependent on post-transcriptional regulation of RNA binding proteins, miRNAs and components of the translational machinery[66]. To counteract the effects of chemotherapeutics, cancer cells take advantage from these mechanisms to regulate the translation rate of selected mRNAs involved in cell survival, EMT, drug metabolism and proliferation. In this context, given their capacity to regulate the stability and translation efficiency of specific mRNAs, P-bodies have a direct role in tumor drug resistance. Protein kinases MEK1/2 act as oncogenes in different types of cancer since they promote cell proliferation and survival, supporting tumor expansion and metastasis. About 30% of human cancers show uncontrolled activation of Ras/Raf/MEK/ERK pathway. Accordingly, potent and selective inhibitors of MEK (MEKi) are currently being used for the treatment of different types of tumors[112,113]. However, many of these tumors develop resistance to the chemotherapeutics and this is mostly associated with high levels of K-Ras and N-Ras proteins[113,114]. P-bodies play a fundamental role in the development of chemoresistance. In fact, mRNAs encoding for both Ras genes are located within P-bodies and in breast, lung and melanoma cancer cells, MEKi treatment induces the resumption of their translation by promoting P-body disassembly. Similar effects are obtained by genetic silencing of DDX6 that promotes P-body dissolution, and expression analysis in cancer tissue samples shows an inverse correlation between DDX6 and Ras protein levels[115]. The molecular mechanism by which MEKi induces P-body disassembly is unclear; however, considering the importance of PTMs in regulating P-body dynamics, we can assume that the MAPK pathway targets essential components of P-bodies, impacting on the dynamics of these mRNP condensates. Indeed, a phosphoproteomic analysis of ERK substrates identified 4E-T as an indirect target of MAPK[116]. Although the mechanisms through which MAPKi leads to drug resistance via P-bodies remain to be addressed, these findings are consistent with the model whereby translational regulation of mRNA at P-bodies in cancer cells controls the expression levels of Ras proteins which, in turn, support chemoresistance[115] (Fig. 3). Therefore, understanding the mechanisms operating at P-bodies that control protein translation represents an important goal for the design of novel therapeutic strategies able to overcome chemoresistance of otherwise aggressive tumors.

## Conclusions and perspectives

P-bodies are RNP condensates that play a fundamental role in the translational control of gene expression. By regulating mRNA storage or decay, P-bodies exert important biological effects underlying key cellular activities. Although P-bodies are constitutively present in cells, the assembly/disassembly of these condensates can be regulated by various signaling pathways and post-translational events that rapidly modify the rate of mRNA translation under different microenvironmental conditions. Cancer cells need to rapidly modify their proteome in response to metabolic changes or modifications of the tumor microenvironment that would otherwise limit cancer cell growth and dissemination. The regulation of P-body dynamics, thus, represents a fundamental mechanism for cancer cells to rapidly change

their translational asset. In the last years, many studies have focused on the role of P-body dynamics in cancer growth and development, demonstrating the relevance of these condensates in cancer cell plasticity. This allows adaptation of proliferating cancer cells to hormone/growth factor stimulation or metabolic needs, promoting expansion of tumor mass and resistance to chemotherapeutics.

The role of P-bodies in regulating tumor growth, metastasis, and response to therapy is a relatively new field, and cumulative evidence supports the existence of a dynamic intersection between these consendates and oncogenic pathways that shapes the proteome of cancer cells. The link between P-bodies and tumors has been established in different types of human cancer. However, it is still debating whether the RNP condensates impact on the behavior of specific types of tumors or they have a more general implication in malignant cell transformation. A rapidly growing tumor is often associated with changes in the levels of RNPs condensates, but these alterations do not always reflect significant modifications of protein translation. Furthermore, the number of P-bodies do not reflect the aggressiveness potential or the type of malignant tumors, since breast cancer growth has been linked to both an increase and a decrease in the number of RNP condensates. Instead, it is most likely a consequence of the highly dynamic nature of P-bodies, which can rapidly assemble/disassemble in response to changes in the tumor microenvironment or nutrient availability. Also, the importance of P-bodies in tumor growth depends on the type of mRNAs they regulate, rather than the abundance of RNPs granules. In fact, although modifications of P-body dynamics in tumor cells are linked to changes of global protein translation, the regulation of specific mRNAs encoding for oncogenes or tumor suppressors by the RNPs condensates is likely more important. As example, studies investigating the role of the hypoxia-P-bodies axis in tumor cells revealed that, under hypoxic conditions, these biomolecular condensates regulate the fate of selected transcripts encoding for proteins involved in cancer cell growth. Similarly, the control of certain transcripts within P-bodies underlies the development of cancer chemoresistance. However, the precise biological role of RNPs condensates in drug resistance, and the molecular mechanisms involved, needs further in-depth investigation.

The current model points to an important role of P-bodies in various aspects of tumor cell biology, such as initiation, proliferation, stemness, development of metastases, stress response, and chemoresistance. However, it is still difficult to determine whether, and on which aspect, they have a greater impact. The main characteristic of P-bodies is the ability to dynamically regulate mRNAs, both through degradation or storage, rapidly and reversibly impacting on protein translation. For this reason, we believe that the role of P-bodies in tumor biology most likely relies on their ability to confer cell plasticity, a characteristic that supports adaptation of cancer cells to rapid changes in the tumor microenvironment, promoting genetic heterogeneity, cell growth, immune evasion, resistance to chemotherapeutics, and metastatic spreading. These aspects are of particular relevance in tumors where no recurrent mutations or molecular alterations driving carcinogenesis have been identified. In this context, P-body dynamics, by conferring plasticity to cancer cells, may contribute to tumor growth and expansion, thus representing relevant therapeutic targets.

In conclusion, this review highlighted the emerging role of P-bodies in cancer. Further studies will elucidate the molecular mechanisms underlying the P-body impact on cancer cell plasticity and will likely contribute to identify and target components and regulators of P-bodies, thus providing novel strategies for the treatment of otherwise aggressive malignant tumors.

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

## Acknowledgements

This work was supported by Fondazione AIRC per la Ricerca sul Cancro (IG2023-29124), the Italian Ministry of University and Research (National Center for Gene Therapy and Drugs based on RNA Technology, PNRR-CN3: E63C22000940007; PRIN2022: E53D23009690006 and E53D23021760001, MNESYS-A multiscale integrated approach to the study of the nervous system in health and disease' PE0000006, DN.1553), European Regional Development Fund (POR Campania FESR 2021-2027), grant "RARE.Glials" to AF. The authors apologize to those whose work was not cited owing to space limitations.

## Author contributions

E.S. and A.F. wrote the article with contributions from L.R., F.C., and A.G.B. E.S. prepared the figures.

## Competing interests

The authors declare no competing interests.
