## [Transparent Peer Review file · Communications Biology]

Reshaping cancer cell plasticity by P-body dynamics and protein translation

Corresponding Author: Professor Antonio Feliciello

Version 0:

Reviewer comments:

Reviewer #1

(Remarks to the Author)

The review is informative but lacks clarity in several sections. Significant improvement can be achieved by making the cancer background sections more concise, and by summarizing the discoveries of key citations more clearly to better connect P-body and cancer literature. Additionally, the authors mostly cited old P-body and stress granule reviews instead of seminal research citations. The seminal P-body and stress granule citations would allow cancer researchers more efficient access to key concepts, players, experimental approaches and challenges to studying condensates.

Introduction comments:

- The first paragraph aims to layout the three levels of gene expression to provide context, however 2/3 levels are not discussed for the rest of the review. More depth can be gained from this space by connecting these three levels with an additional theme. Two ways to connect these three levels are localization and to acknowledge that different types of condensates function at each level. This would provide readers with a broader framework that allows for focus on cytoplasmic regulation, while providing the citations for condensates that affect gene expression in the nucleus. A figure summarizing these concepts would be useful.
- Lines 59-62: Provide seminal citations as there are not many of them. Citing another P-bodies in cancer review from 2022 is not helpful.

P-bodies properties and biological functions:

- Lines 70-72: Citation 4 is for a review on post-transcriptional control of gene expression in cancer. Since the subject of the sentence are condensates, please cite a review on condensates or cytoplasmic condensates.
- Lines 72-75: Citations 10, 11 are both reviews. A strong but old 2014 review on condensates and a newer review on phase separation and cancer. Provide 3-4 seminal citations that demonstrate the localization and condensation of mRNAs in condensates.
- Lines 84-86: Citation 17 is for drosophila cells. Although many condensates are evolutionarily conserved, their functions and compositions can differ from insects to humans. For this review, which aims to connect P-bodies to human cancers, it is best if the background information is based on mammalian P-bodies and stress granules.
- Lines 99-100: The authors state that the assembly of P-bodies requires the core component EDC3, however one of the cited sources Ayache et al. (25) states that silencing EDC3 does not impact P-bodies and leads to an increase in P-body number.
- Lines 102-110: This section blurs the boundaries between what is known about yeast P-bodies and what is known about human P-bodies. This is an issue because, although there are some overlaps, P-body interaction networks and their responses to different types of stress differ significantly between yeast and mammals. For example, the yeast decapping complex interaction network is different from the mammalian decapping complex network despite having similar players. Also, yeast P-bodies are strongly responsive to glucose starvation, whereas mammalian P-bodies are not. Heat shock stress represents an evolutionarily conserved P-body response in yeast, insects and mammals. This is a critical section for the authors to establish a solid foundation about mammalian P-bodies for the readers to fully appreciate the following sections. The authors should focus is on what is known about mammalian P-bodies, such as what are their key interactions, what types of stress have been shown to trigger their formation, disassembly and rewiring. If cancer relevant molecules and pathways have been linked to yeast P-bodies but haven't been linked to mammalian P-bodies yet, then the authors would best serve the audience by identifying this relationship and mentioning that it has not been tested in mammalian cells. The transparency and clarity of this section will allow readers to develop the rationale,

framework and enthusiasm for evaluating the connections of mammalian P-bodies to cancer relevant molecules and pathways.

Lines 116-133: This paragraph weaves between yeast and mammals a little more clearly, however yeast studies, which haven't been tested in mammals, can serve more clarity when connected to cancer-specific areas in the later sections.

Control of P-body dynamics by post-translational modifications:

- Lines 135-139: Provide updated references. Ideally with seminal citations to mammalian studies and a more recent review on P-bodies, which may or may not include stress granules and other condensates.

- Lines 140-143: Citation 38, although relevant to DDX3X, does not test or mention P-bodies at all. DDX3X is a stress granule protein and is not enriched in P-bodies.

Lines 140-151: This is a confusing paragraph. Also, what is the effect of ablating the ATPase activity of human DDX6 on P-bodies? Only yeast Dhh1 (DDX6) is mentioned and cited.

- Lines 152-167: Good summary of phosphorylation and ubiquitination targets/consequences.

Lines 160-167: This section successfully outlined the differences and similarities between yeast and mammals in the relationship between PKA signaling and P-bodies.

A role of P-bodies in cancer development and progression:

- Lines 169-177: Make this section more concise. Key points are mRNA translation and protein synthesis alterations contribute to cancer cell plasticity by altering the expression of “oncogenes, tumor suppressors and growth regulators.” Stress granules and P-bodies are two cytoplasmic condensates that can impact the translation of specific mRNAs. Are these two condensates working together in different cancers or are they antagonistic of each other?

- Lines 177-183: Summarize the key discoveries of each citation for stress granules (53-58) and P-bodies (59 and 60) similar to how citations 61-63 were summarized.

- Lines 188-193: HER2+ breast cancer cells co-overexpressing both MLN51 and HER2 displayed fewer P-bodies than breast cancer cells just HER2 overexpression. Here, can the authors clarify whether the disassembly of P-bodies is correlated with increased tumor aggressiveness and patient mortality?

Regulation of P-body dynamics by signaling pathways in cancer:

- Lines 207-214: Nice summary of citation 65. Are there other seminal papers identifying EDC3 phosphorylation by PIM kinase? If so, then please include.

Lines 207-208: Please clarify what “altered P-body structure” means. Are the biophysical properties different demonstrated potential through a FRAP assay? Or are the number and size of P-bodies affected by reduced P-body enrichment of S161-phosphorylated EDC3? If it's the latter, then “altered P-body structure” might be an overinterpretation of the results.

- Lines 214-222: Nice summary of TGFB signaling and P-body connections.

- Lines 207-222: This paragraph includes PIM kinase signaling and TGFB signaling. Do P-bodies connect these pathways in some way? Does TGFB signaling lead to the upregulation of PIM kinases? Does PIM kinase-dependent phosphorylation of EDC3 enhance or dampen TGFB signaling?

- Lines 239-241: Which citation (74-79) corresponds to PKA-dependent ubiquitination of DDX6?

- Lines 242-244: Describe how DDX6 ubiquitination was disrupted? Point mutations? Or domain deletion?

P-bodies, hypoxia and metabolic reprogramming in cancer:

- Lines 248-264: What is the message of the first half of this paragraph? That cancer cells have a low levels of ATP, and that low levels of ATP were shown to induce P-bodies in CRC cells? This paragraph can be included in the “Regulation of P-body dynamics by signaling pathways in cancer” section.

Lines 257-260: Citation 86 needs to be cited these sentences. Citations 84, 85 only identify that HAX1 is enriched in P-bodies, however they do not mention ATP levels, Trim23 signaling, etc.

- Lines 265-276: Do P-bodies from mammalian cells respond to hypoxia? Also, the exosome is excluded from P-bodies and the paper cited does not show EXOSC9 enrichment inside P-bodies. Thus, there does not appear to be a direct link between hypoxia and P-body formation. If the authors want to include a hypoxia section, then it would be useful to identify the knowledge gap and provide citations that might make the P-body/hypoxia relationship an attractive target.

A role of P-bodies in cancer stemness:

- Important connection. The authors clearly linked P-body and ADAR1 function through citations showing that CPEB3 regulates ADAR1 protein levels by recruiting and repressing ADAR1 mRNA in P-bodies. Since ADAR1 protein has also been shown to localize to P-bodies, please clarify with citations where ADAR1-dependent mRNA editing is occurring? Is it in the nucleus, cytoplasm and/or P-bodies?

Figures

- Figure 1a: This figure illustrates the ubiquitylation of Dcp1A, stress-induced phosphorylation of 4-ET, ubiquitylation of 4-ET, and de-methylated state of Lsm4 – all conditions which lead to P-body assembly. However, the phosphorylation of serine 315 on Dcp1A by Jnk (Line 154) is not depicted here.

- Figure 1b: Tenekeci et al. (43) states that mutating C-terminal residues of Dcp1A, which are targeted for ubiquitination, results in a loss of interaction with Dcp2, EDC4, and Xrn1, but not EDC3. What is the origin of the loss of interaction between Dcp1A and EDC3 in figure 1B?

- Figure 2: This figure clearly illustrates specific examples of the role of P-body assembly/disassembly in cancer.

Is Trim23 ubiquitylation of HAX1 specific to stress induced by low ATP or can this be induced by alternative sources of cellular stress?

In prostate cancer cells, is EDC3 phosphorylation by PIM1/3 kinases promoting the disassembly of P-bodies or is it preventing the assembly of P-bodies?

Reviewer #2

(Remarks to the Author)

I co-reviewed this manuscript with one of the reviewers who provided the listed reports. This is part of the Communications Biology initiative to facilitate training in peer review and to provide appropriate recognition for Early Career Researchers who co-review manuscripts.

Reviewer #3

(Remarks to the Author)

General Assessment

This manuscript addresses a timely and significant topic in cancer biology by exploring the role of P-bodies in shaping cancer cell plasticity. The deliberate exclusion of stress granules allows for a more focused narrative, which could potentially bring conceptual clarity to an otherwise complex area. The authors aim to synthesize a growing body of literature on RNA granule dynamics in cancer, and in principle, this is a commendable objective. However, in its current form, the manuscript remains largely descriptive and does not yet provide the depth of analysis, critical insight, or integrative vision expected of a high-level scholarly review.

Major Concerns

Lack of Critical Perspective and Integrative Analysis

The review primarily catalogs individual studies without offering an overarching analytical framework. Readers are left without a clear sense of the strength or hierarchy of the evidence presented, or of the comparative significance of P-bodies relative to other RNA granules in cancer biology. The review does not clearly articulate whether P-bodies contribute meaningfully to oncogenesis, progression, or treatment resistance, nor does it establish in which cancer types their role is best supported.

The absence of evaluative synthesis results in a missed opportunity to clarify conceptual advances and identify critical knowledge gaps. The authors, given their expertise, are well positioned to guide the field by assessing the consistency and mechanistic depth of the available literature. The manuscript would benefit from a dedicated section that distills the most salient findings, highlights conflicting data where relevant, and articulates a reasoned expert perspective on how P-body biology might intersect with future therapeutic strategies.

Conceptual Confusion Between Protein Expression Changes and Functional Alterations in P-bodies

A persistent issue throughout the text is the conflation of protein-level changes with changes in P-body behavior or function. Many studies are cited that report modulation of P-body-associated proteins, but it remains unclear whether these changes translate into modifications in P-body number, size, composition, or regulatory activity.

This distinction is essential. An increase or decrease in the expression of a P-body component does not necessarily imply a change in P-body-mediated mRNA regulation. The manuscript should make it clear whether reported protein alterations impact granule formation, RNA stability, translational repression, or other measurable outcomes. Clarifying this mechanistic chain would substantially improve the review's coherence and scientific rigor. Without such clarification, the review risks presenting correlative findings as causal mechanisms, which undermines its interpretive value.

Incomplete Coverage of Recent Literature

Given the fast-evolving nature of RNA granule biology, particularly in cancer contexts, the literature review must be exhaustive and up to date. Several key studies published between 2023 and 2025 are noticeably absent, despite their relevance. The omission of these works not only weakens the manuscript's authority as a reference text but also risks presenting an outdated view of P-body function.

The field has progressed significantly since Hubstenberger et al. (*Mol Cell*, 2007), which repositioned P-bodies as reservoirs of stable mRNAs rather than decay sites. Statements in the current manuscript that refer to P-bodies primarily as sites of RNA degradation may confuse readers who are not familiar with the evolution of this concept.

For instance, the recent work by Safieddine et al. (*Mol Cell*, 2024) has demonstrated cell cycle-regulated localization of mRNAs within P-bodies, a finding with clear implications for cancer cell proliferation. Other foundational studies that link P-bodies to proliferation, such as those by Yang et al. (2004) and Lian et al. (2006), should be more thoroughly integrated. The authors are encouraged to conduct a systematic literature search to ensure that all relevant contributions are represented and critically appraised.

Safieddine A, Benassy MN, Bonte T, Slimani F, Pourcelot O, Kress M, Ernoult-Lange M, Courel M, Coleno E, Imbert A, Laine A, Godebert AM, Vinit A, Blugeon C, Chevreux G, Gautheret D, Walter T, Bertrand E, Bénard M, Weil D. Cell-cycle-dependent mRNA localization in P-bodies. *Mol Cell*. 2024 Nov 7;84(21):4191-4208.e7. doi: 10.1016/j.molcel.2024.09.011. Epub 2024 Oct 4. PMID: 39368464.

Yang Z, Jakymiw A, Wood MR, Eystathioy T, Rubin RL, Fritzier MJ, Chan EK. GW182 is critical for the stability of GW bodies expressed during the cell cycle and cell proliferation. *J Cell Sci*. 2004 Nov 1;117(Pt 23):5567-78. doi: 10.1242/jcs.01477. Epub 2004 Oct 19. PMID: 15494374.

Lian S, Jakymiw A, Eystathioy T, Hamel JC, Fritzier MJ, Chan EK. GW bodies, microRNAs and the cell cycle. *Cell Cycle*. 2006 Feb;5(3):242-5. doi: 10.4161/cc.5.3.2410. Epub 2006 Feb 14. PMID: 16418578.

Underdeveloped Discussion of Functional and Therapeutic Relevance

While the review touches on key aspects such as cancer stemness, metastasis, and chemoresistance, these sections are

insufficiently developed. For example, the reference to MEK inhibitor–induced P-body disassembly in breast, lung, and melanoma cells is particularly intriguing, yet the mechanistic and therapeutic implications are not adequately explored. Similarly, the section on cancer stemness is disproportionately brief. Only one example is discussed in detail, and the following paragraph largely repeats the previous conclusion without expanding on it or offering new insights. The reference to ADAR1 lacks proper citation and explanation. This section, in particular, would benefit from additional examples and a more critical analysis of whether P-body dynamics affect self-renewal pathways or differentiation cues.

Minor Issues

- The manuscript often discusses general features of P-bodies rather than their dynamic regulation specifically in cancer contexts, which somewhat dilutes its central thesis.
- Findings from yeast studies should be summarized more concisely and reframed to emphasize relevance to mammalian systems.
- At line 99, the authors should include LSM14A among the canonical P-body components.
- Certain sentences require grammatical refinement and attention to tense consistency. For instance, lines 220–222 read: “Thus, in orthotopic models of breast cancer, downregulation of DDX6 prevents the assembly of P-bodies and inhibited the formation of pulmonary metastasis,” which combines past and present tense incorrectly.
- The term “organelles” is repeatedly used to describe P-bodies. In the recent literature, these structures are more accurately referred to as “cytoplasmic ribonucleoprotein granules” or “mRNP aggregates.” For terminological consistency and precision, the authors should adopt language that aligns with current usage.
- Some statements are overly brief and isolated. For example, line 230 states, “The availability of mRNA is essential for the formation of P-bodies,” without linking it to broader functional implications.
- Content between lines 248 and 264 could be coupled with earlier part (lines 140 to 152) to improve structural coherence.
- Several sections, particularly those addressing chemoresistance, would benefit from clearer mechanistic exposition and stronger evidence integration.

Conclusion

This manuscript touches on a topic of clear importance and offers the foundation for a valuable contribution to the field. However, significant revisions are required to elevate the review from a descriptive summary to a critical and conceptually mature synthesis. With deeper analysis, stronger integration of recent work, and greater attention to mechanistic clarity, the review could become a central reference in this area of research.

Recommendation: Major revision necessary before the manuscript can be considered for publication.

Version 1:

Reviewer comments:

Reviewer #1

(Remarks to the Author)

The revised review is improved. For clarity, below are suggestions to increase readability. The review can also benefit from broader mentions of the big questions in cancer and how P-bodies can be the key to answering these question.

Introduction: Improved and more clearly demonstrating that condensates are operating at each level.

P-bodies properties and biological functions: This is an important section for readers to connect with the rest of the sections. It is disjointed with a bunch of condensates that are not related or somewhat related to P-bodies. My suggestion is to split the section into smaller more purposeful sections:

1. Membrane-less organelles and phase separation: Short section laying out the concept underlying phase separation and membrane-less organelles including the no-P-body examples mentioned, as well as species specific condensates that are similar to P-bodies, but not P-bodies (GW bodies, P-granules, etc). You can end it with the P-body and stress granule verbiage in the first paragraph.
2. P-bodies background and historical context: Start with paragraph 2 of “P-bodies properties and biological functions” section, and gather any critical yeast information here. Note: In yeast, under homeostasis (log-phase growth), P-bodies aren’t present, which is more similar to human stress granules. Human P-bodies are constitutively present. This section can finish with something similar to these two points separated in your current introduction: “In yeast, P-bodies assembly is primed by different environmental conditions, such as glucose deprivation, osmotic stress and DNA replication stress, but these events have not been confirmed in all eukaryotes. Since most of these studies have been made in yeast cells and the findings might not be extrapolated to mammalian systems, this review will primarily focus on the structure and regulation of mammalian P-bodies.”
3. Key players of human P-bodies: Start with paragraph 3 of “P-bodies properties and biological functions” section and finish the section with human aspects of paragraphs 4 and 5. Maybe move the post-translational information into your “Control of P-body dynamics by post-translational modifications.” section?

Control of P-body dynamics by post-translational modifications: Useful section. Since this a general P-body PTM section, the next section can have cancer-related P-body PTM-containing sections like the “Regulation of P-body dynamics by signaling pathways in cancer” section.

A role of P-bodies in cancer development and progression: Improved.

Regulation of P-body dynamics by signaling pathways in cancer: No comments other than potentially positioning this section after the "Control of P-body dynamics by post-translational modifications" section above.

P-bodies, hypoxia and metabolic reprogramming in cancer: Improved.

A role of P-bodies in cancer stemness: Improved.

P-body in cancer chemoresistance: Improved.

Conclusions and perspectives: Improved.

Reviewer #2

(Remarks to the Author)

I co-reviewed this manuscript with one of the reviewers who provided the listed reports. This is part of the Communications Biology initiative to facilitate training in peer review and to provide appropriate recognition for Early Career Researchers who co-review manuscripts.

Reviewer #3

(Remarks to the Author)

The authors have responded to all the points raised and the article is acceptable as it stands.

Version 2:

Reviewer comments:

Reviewer #1

(Remarks to the Author)

No further comments, the authors addressed all comments.

Reviewer #1 (Remarks to the Author):

The review is informative but lacks clarity in several sections. Significant improvement can be achieved by making the cancer background sections more concise, and by summarizing the discoveries of key citations more clearly to better connect P-body and cancer literature. Additionally, the authors mostly cited old P-body and stress granule reviews instead of seminal research citations. The seminal P-body and stress granule citations would allow cancer researchers more efficient access to key concepts, players, experimental approaches and challenges to studying condensates.

R. *We wish to thank the reviewer for the insightful comments that we have now fully addressed as suggested.*

Introduction comments:

- The first paragraph aims to layout the three levels of gene expression to provide context, however 2/3 levels are not discussed for the rest of the review. More depth can be gained from this space by connecting these three levels with an additional theme. Two ways to connect these three levels are localization and to acknowledge that different types of condensates function at each level. This would provide readers with a broader framework that allows for focus on cytoplasmic regulation, while providing the citations for condensates that affect gene expression in the nucleus. A figure summarizing these concepts would be useful.

R. *We have now discussed the three levels of regulation of gene expression including a new schematic figure of. (pp.2, Lines 40-59, new Figure 1)*

- Lines 59-62: Provide seminal citations as there are not many of them. Citing another P-bodies in cancer review from 2022 is not helpful.

R. *We included new references (pp.2, line 64, ref. 10-14)*

P-bodies properties and biological functions:

- Lines 70-72: Citation 4 is for a review on post-transcriptional control of gene expression in cancer. Since the subject of the sentence are condensates, please cite a review on condensates or cytoplasmic condensates.

R. *We have now cited a review on membraneless condensates (pp.3, line 74, ref. 16)*

- Lines 72-75: Citations 10, 11 are both reviews. A strong but old 2014 review on condensates and a newer review on phase separation and cancer. Provide 3-4 seminal citations that demonstrate the localization and condensation of mRNAs in condensates.

R. *As suggested, we included new citations (pp.3, line 77, ref. 17-21)*

- Lines 84-86: Citation 17 is for drosophila cells. Although many condensates are evolutionarily conserved, their functions and compositions can differ from insects to humans. For this review, which aims to connect P-bodies to human cancers, it is best if the background information is based on mammalian P-bodies and stress granules.

R. *Many thanks for the comment. We have now included appropriate references on mammalian U-bodies (pp.3, line 88, ref. 29)*

- Lines 99-100: The authors state that the assembly of P-bodies requires the core

component EDC3, however one of the cited sources Ayache et al. (25) states that silencing EDC3 does not impact P-bodies and leads to an increase in P-body number.

R. *Very helpful comment. We corrected the statement, since EDC3 is not essential for P-body assembly (pp.4, lines 111-112).*

- Lines 102-110: This section blurs the boundaries between what is known about yeast P-bodies and what is known about human P-bodies. This is an issue because, although there are some overlaps, P-body interaction networks and their responses to different types of stress differ significantly between yeast and mammals. For example, the yeast decapping complex interaction network is different from the mammalian decapping complex network despite having similar players. Also, yeast P-bodies are strongly responsive to glucose starvation, whereas mammalian P-bodies are not. Heat shock stress represents an evolutionarily conserved P-body response in yeast, insects and mammals.

This is a critical section for the authors to establish a solid foundation about mammalian P-bodies for the readers to fully appreciate the following sections. The authors should focus is on what is known about mammalian P-bodies, such as what are their key interactions, what types of stress have been shown to trigger their formation, disassembly and rewiring. If cancer relevant molecules and pathways have been linked to yeast P-bodies but haven't been linked to mammalian P-bodies yet, then the authors would best serve the audience by identifying this relationship and mentioning that it has not been tested in mammalian cells. The transparency and clarity of this section will allow readers to develop the rationale, framework and enthusiasm for evaluating the connections of mammalian P-bodies to cancer relevant molecules and pathways.

R. *We have now improved this section. Although several key discoveries about P-bodies were initially made in yeast, we have now shifted our focus to studies in mammals. Yeast studies are mentioned when relevant, and in all cases, we clearly indicate the organism studied (pp. 4-5)*

Lines 116-133: This paragraph weaves between yeast and mammals a little more clearly, however yeast studies, which haven't been tested in mammals, can serve more clarity when connected to cancer-specific areas in the later sections.

R. *As suggested, we have now clarified if studies in yeast were confirmed in mammals (pp.5, lines 163-164)*

Control of P-body dynamics by post-translational modifications:

- Lines 135-139: Provide updated references. Ideally with seminal citations to mammalian studies and a more recent review on P-bodies, which may or may not include stress granules and other condensates.

R. *Many thanks for this comment. We removed that statement.*

- Lines 140-143: Citation 38, although relevant to DDX3X, does not test or mention P-bodies at all. DDX3X is a stress granule protein and is not enriched in P-bodies.

Lines 140-151: This is a confusing paragraph. Also, what is the effect of ablating the ATPase activity of human DDX6 on P-bodies? Only yeast Dhh1 (DDX6) is mentioned and cited.

R. *As suggested, we decided to remove this study.*

- Lines 152-167: Good summary of phosphorylation and ubiquitination targets/consequences.

Lines 160-167: This section successfully outlined the differences and similarities between yeast and mammals in the relationship between PKA signaling and P-bodies.

R. *Many thanks for the positive comment.*

A role of P-bodies in cancer development and progression:

- Lines 169-177: Make this section more concise. Key points are mRNA translation and protein synthesis alterations contribute to cancer cell plasticity by altering the expression of “oncogenes, tumor suppressors and growth regulators.” Stress granules and P-bodies are two cytoplasmic condensates that can impact the translation of specific mRNAs. Are these two condensates working together in different cancers or are they antagonistic of each other?

R. *As suggested, we have now summarized the beginning of the paragraph, and focused exclusively on P-bodies to improve clarity (pp. 6, lines 192-198).*

- Lines 177-183: Summarize the key discoveries of each citation for stress granules (53-58) and P-bodies (59 and 60) similar to how citations 61-63 were summarized.

R. *As also noted by the referees, the review specifically focuses on the role of P-bodies in cancer. Accordingly, we removed the references related to the role of stress granules in tumor progression.*

- Lines 188-193: HER2+ breast cancer cells co-overexpressing both MLN51 and HER2 displayed fewer P-bodies than breast cancer cells just HER2 overexpression. Here, can the authors clarify whether the disassembly of P-bodies is correlated with increased tumor aggressiveness and patient mortality?

R. *We have now clarified this point (pp. 7, lines 209-211).*

Regulation of P-body dynamics by signaling pathways in cancer:

- Lines 207-214: Nice summary of citation 65. Are there other seminal papers identifying EDC3 phosphorylation by PIM kinase? If so, then please include.

R. *We have now cited seminal papers identifying EDC3 phosphorylation by PIM kinase. (pp. 7, lines 228, ref. 71-73).*

Lines 207-208: Please clarify what “altered P-body structure” means. Are the biophysical properties different demonstrated potential through a FRAP assay? Or are the number and size of P-bodies affected by reduced P-body enrichment of S161-phosphorylated EDC3? If it's the latter, then “altered P-body structure” might be an overinterpretation of the results.

R. *Thanks for your comment. We have now clarified that phosphorylation affects EDC3 localization. (pp. 8, lines 228-229).*

- Lines 214-222: Nice summary of TGFB signaling and P-body connections.

R. *Many thanks for the positive comment.*

- Lines 207-222: This paragraph includes PIM kinase signaling and TGFB signaling. Do P-bodies connect these pathways in some way? Does TGFB signaling lead to the upregulation of PIM kinases? Does PIM kinase-dependent phosphorylation of EDC3 enhance or dampen TGFB signaling?

R. *Currently, we have not found studies that report a direct correlation between TGF- β and Pim kinase in the regulation of P-bodies and cancer. However, it is plausible that there are connections between these signaling pathways. We have chosen to keep separate the descriptions of the mechanisms involved in P-body dynamics in different tumor contexts (pp.7-8, lines 227-243).*

• Lines 239-241: Which citation (74-79) corresponds to PKA-dependent ubiquitination of DDX6?

R. *We have now indicated the specific citation. (pp. 8, line 264, ref. 64).*

• Lines 242-244: Describe how DDX6 ubiquitination was disrupted? Point mutations? Or domain deletion?

R. *We have now indicated that it is a point mutant. (pp. 8, line 263)*

P-bodies, hypoxia and metabolic reprogramming in cancer:

• Lines 248-264: What is the message of the first half of this paragraph? That cancer cells have a low levels of ATP, and that low levels of ATP were shown to induce P-bodies in CRC cells? This paragraph can be included in the “Regulation of P-body dynamics by signaling pathways in cancer” section.

R. *As suggested, we moved the paragraph to the section “Regulation of P-body dynamics by signaling pathways in cancer” and we have now better explained the role of ATP in P-bodies dynamics and cancer. (pp.8-9, lines 267-275).*

Lines 257-260: Citation 86 needs to be cited these sentences. Citations 84, 85 only identify that HAX1 is enriched in P-bodies, however they do not mention ATP levels, Trim23 signaling, etc.

R. *We have now indicated the correct reference (pp.9, line 274, ref. 49).*

• Lines 265-276: Do P-bodies from mammalian cells respond to hypoxia? Also, the exosome is excluded from P-bodies and the paper cited does not show EXOSC9 enrichment inside P-bodies. Thus, there does not appear to be a direct link between hypoxia and P-body formation. If the authors want to include a hypoxia section, then it would be useful to identify the knowledge gap and provide citations that might make the P-body/hypoxia relationship an attractive target.

R. *Many thanks for the comment. EXOSC9 controls the assembly of P-bodies although it is not a resident protein of these condensates. We have now better described the cited study (pp.10, lines 317-327). Moreover, we have now included other studies suggesting a relation between P-bodies and hypoxia, even if we clearly indicate that a direct link has still not well established (pp. 9 -10, lines 277-316, ref. 92-95)*

A role of P-bodies in cancer stemness:

• Important connection. The authors clearly linked P-body and ADAR1 function through citations showing that CPEB3 regulates ADAR1 protein levels by recruiting and repressing ADAR1 mRNA in P-bodies. Since ADAR1 protein has also been shown to localize to P-bodies, please clarify with citations where ADAR1-dependent mRNA editing is occurring? Is it in the nucleus, cytoplasm and/or P-bodies?

R. *Many thanks for the observation. We have now better clarified this point. (pp.11, lines 338-340, ref. 99)*

Figures

• Figure 1a: This figure illustrates the ubiquitylation of Dcp1A, stress-induced phosphorylation of 4-ET, ubiquitylation of 4-ET, and de-methylated state of Lsm4 – all conditions which lead to P-body assembly. However, the phosphorylation of serine 315 on Dcp1A by Jnk (Line 154) is not depicted here.

R. *As suggested, we have now included the Dcp1A phosphorylation by Jnk in the figure (pp. 16, line 487, figure 2A)*

• Figure 1b: Tenekeci et al. (43) states that mutating C-terminal residues of Dcp1A, which are targeted for ubiquitination, results in a loss of interaction with Dcp2, EDC4, and Xrn1, but not EDC3. What is the origin of the loss of interaction between Dcp1A and EDC3 in figure 1B?

R. *We have corrected the figure, maintaining the interaction between Dcp1A and EDC3 (figure 2B)*

• Figure 2: This figure clearly illustrates specific examples of the role of P-body assembly/disassembly in cancer. Is Trim23 ubiquitylation of HAX1 specific to stress induced by low ATP or can this be induced by alternative sources of cellular stress? In prostate cancer cells, is EDC3 phosphorylation by PIM1/3 kinases promoting the disassembly of P-bodies or is it preventing the assembly of P-bodies?

R. *Thanks for the helpful comment. HAX1 ubiquitylation is specific to energy stress and we have now corrected the figure. We have also indicated that EDC3 phosphorylation inhibits the assembly of P-bodies (pp. 16, lines 503 and 508, figure 3)*

Reviewer #2 (Remarks to the Author):

I co-reviewed this manuscript with one of the reviewers who provided the listed reports. This is part of the Communications Biology initiative to facilitate training in peer review and to provide appropriate recognition for Early Career Researchers who co-review manuscripts.

Reviewer #3 (Remarks to the Author):

General Assessment

This manuscript addresses a timely and significant topic in cancer biology by exploring the role of P-bodies in shaping cancer cell plasticity. The deliberate exclusion of stress granules allows for a more focused narrative, which could potentially bring conceptual clarity to an otherwise complex area. The authors aim to synthesize a growing body of literature on RNA granule dynamics in cancer, and in principle, this is a commendable objective. However, in its current form, the manuscript remains largely descriptive and does not yet provide the depth of analysis, critical insight, or integrative vision expected of a high-level scholarly review.

R. *We wish to thank the reviewer for the positive comment on the objective of the review and for the insightful comments that we have carefully addressed as suggested.*

Major Concerns

Lack of Critical Perspective and Integrative Analysis

The review primarily catalogs individual studies without offering an overarching analytical framework. Readers are left without a clear sense of the strength or hierarchy of the evidence presented, or of the comparative significance of P-bodies relative to other RNA granules in cancer biology. The review does not clearly articulate whether P-bodies contribute meaningfully to oncogenesis, progression, or treatment resistance, nor does it establish in which cancer types their role is best supported.

The absence of evaluative synthesis results in a missed opportunity to clarify conceptual

advances and identify critical knowledge gaps. The authors, given their expertise, are well positioned to guide the field by assessing the consistency and mechanistic depth of the available literature. The manuscript would benefit from a dedicated section that distills the most salient findings, highlights conflicting data where relevant, and articulates a reasoned expert perspective on how P-body biology might intersect with future therapeutic strategies.

Conceptual Confusion Between Protein Expression Changes and Functional Alterations in P-bodies

A persistent issue throughout the text is the conflation of protein-level changes with changes in P-body behavior or function. Many studies are cited that report modulation of P-body-associated proteins, but it remains unclear whether these changes translate into modifications in P-body number, size, composition, or regulatory activity.

This distinction is essential. An increase or decrease in the expression of a P-body component does not necessarily imply a change in P-body-mediated mRNA regulation.

The manuscript should make it clear whether reported protein alterations impact granule formation, RNA stability, translational repression, or other measurable outcomes.

Clarifying this mechanistic chain would substantially improve the review's coherence and scientific rigor. Without such clarification, the review risks presenting correlative findings as causal mechanisms, which undermines its interpretive value.

R. *Many thanks for these comments. We have now improved the section of conclusions and perspectives including a more general view of the published work, knowledge gaps and critical view regarding the most relevant discoveries on the role of P-bodies in cancer (pp. 13-14, lines 418-451). Also, we have clarified, throughout the review, if modulation of levels of P-bodies components affect the number or activity of the condensates and whether alterations of P-bodies correlate with modifications of protein translation.*

Incomplete Coverage of Recent Literature

Given the fast-evolving nature of RNA granule biology, particularly in cancer contexts, the literature review must be exhaustive and up to date. Several key studies published between 2023 and 2025 are noticeably absent, despite their relevance. The omission of these works not only weakens the manuscript's authority as a reference text but also risks presenting an outdated view of P-body function.

The field has progressed significantly since Hubstenberger et al. (Mol Cell, 2007), which repositioned P-bodies as reservoirs of stable mRNAs rather than decay sites. Statements in the current manuscript that refer to P-bodies primarily as sites of RNA degradation may confuse readers who are not familiar with the evolution of this concept.

For instance, the recent work by Safieddine et al. (Mol Cell, 2024) has demonstrated cell cycle-regulated localization of mRNAs within P-bodies, a finding with clear implications for cancer cell proliferation. Other foundational studies that link P-bodies to proliferation, such as those by Yang et al. (2004) and Lian et al. (2006), should be more thoroughly integrated. The authors are encouraged to conduct a systematic literature search to ensure that all relevant contributions are represented and critically appraised.

Safieddine A, Benassy MN, Bonte T, Slimani F, Pourcelot O, Kress M, Ernoult-Lange M, Courel M, Coleno E, Imbert A, Laine A, Godebert AM, Vinit A, Blugeon C, Chevreaux G, Gautheret D, Walter T, Bertrand E, Bénard M, Weil D. Cell-cycle-dependent mRNA localization in P-bodies. Mol Cell. 2024 Nov 7;84(21):4191-4208.e7. doi: 10.1016/j.molcel.2024.09.011. Epub 2024 Oct 4. PMID: 39368464.

Yang Z, Jakymiw A, Wood MR, Eystathioy T, Rubin RL, Fritzler MJ, Chan EK. GW182 is

critical for the stability of GW bodies expressed during the cell cycle and cell proliferation. J Cell Sci. 2004 Nov 1;117(Pt 23):5567-78. doi: 10.1242/jcs.01477. Epub 2004 Oct 19.

PMID: 15494374.

Lian S, Jakymiw A, Eystathioy T, Hamel JC, Fritzler MJ, Chan EK. GW bodies, microRNAs and the cell cycle. *Cell Cycle*. 2006 Feb;5(3):242-5. doi: 10.4161/cc.5.3.2410. Epub 2006 Feb 14. PMID: 16418578.

R. *Many thanks for the helpful comments/suggestions. We have now improved the cited studies by including new relevant studies along with the suggested references (ref. 11, 30, 31)*

Underdeveloped Discussion of Functional and Therapeutic Relevance

While the review touches on key aspects such as cancer stemness, metastasis, and chemoresistance, these sections are insufficiently developed. For example, the reference to MEK inhibitor–induced P-body disassembly in breast, lung, and melanoma cells is particularly intriguing, yet the mechanistic and therapeutic implications are not adequately explored. Similarly, the section on cancer stemness is disproportionately brief. Only one example is discussed in detail, and the following paragraph largely repeats the previous conclusion without expanding on it or offering new insights. The reference to ADAR1 lacks proper citation and explanation. This section, in particular, would benefit from additional examples and a more critical analysis of whether P-body dynamics affect self-renewal pathways or differentiation cues.

R. *As suggested, we have now better discussed the relation between MEKi and P-body (pp. 12, lines 392-398). Also, we have improved the description and discussion of ADAR1 references (pp. 11, lines 338-350) and included new studies regarding the role of P-bodies in cancer stemness (pp. 11, lines 357-365, ref. 108-109). Furthermore, we added a comment on this issue (pp.11, lines 365-369)*

Minor Issues

•The manuscript often discusses general features of P-bodies rather than their dynamic regulation specifically in cancer contexts, which somewhat dilutes its central thesis.

R. *We thank the reviewer for this comment. We understand the concern, but we respectfully believe that including an overview of general P-body biology is essential for contextualizing their roles in cancer. P-body dynamics is regulated by different mechanisms, many of which have been only recently related to cancer. Accordingly, we think that the introduction is necessary in order to better appreciate the relevance of P-bodies in oncogenesis.*

•Findings from yeast studies should be summarized more concisely and reframed to emphasize relevance to mammalian systems.

R. *We have now focused the attention on the description of mammalian p-bodies. When needed, studies conducted in yeast have been cited, always clarifying the organism under study (pp.4, lines 108-110; pp.5)*

•At line 99, the authors should include LSM14A among the canonical P-body components.

R. *As suggested, we have included LSM14A as component of P-bodies (pp. 4, line 112).*

•Certain sentences require grammatical refinement and attention to tense consistency. For instance, lines 220–222 read: “Thus, in orthotopic models of breast cancer, downregulation of DDX6 prevents the assembly of P-bodies and inhibited the formation of pulmonary metastasis,” which combines past and present tense incorrectly.

R. *Many thanks for the comment. We have corrected this sentence (pp. 8, lines 263-268) as well as other parts of the text*

•The term “organelles” is repeatedly used to describe P-bodies. In the recent literature, these structures are more accurately referred to as “cytoplasmic ribonucleoprotein granules” or “mRNP aggregates.” For terminological consistency and precision, the authors should adopt language that aligns with current usage.

R. *Very helpful comment. We have now used the correct terms throughout the manuscript.*

•Some statements are overly brief and isolated. For example, line 230 states, “The availability of mRNA is essential for the formation of P-bodies,” without linking it to broader functional implications.

R. *We have now corrected this and other isolated periods in the manuscript (pp. 8, lines 252-255).*

•Content between lines 248 and 264 could be coupled with earlier part (lines 140 to 152) to improve structural coherence.

R. *We moved this paragraph to the previous section (pp. 8, lines 267-275).*

•Several sections, particularly those addressing chemoresistance, would benefit from clearer mechanistic exposition and stronger evidence integration.

R. *This is an important point. The role of P-bodies in the development of chemoresistance is a very new field of study, so there are not many studies that support it. Nevertheless, we believe it is a highly important topic, which is why we have decided to discuss it, adding a critical perspective (pp. 12, lines 393-403).*

Conclusion

This manuscript touches on a topic of clear importance and offers the foundation for a valuable contribution to the field. However, significant revisions are required to elevate the review from a descriptive summary to a critical and conceptually mature synthesis. With deeper analysis, stronger integration of recent work, and greater attention to mechanistic clarity, the review could become a central reference in this area of research.

Recommendation: Major revision necessary before the manuscript can be considered for publication.

Reviewers' comments:

Reviewer #1 (Remarks to the Author):

The revised review is improved. For clarity, below are suggestions to increase readability. The review can also benefit from broader mentions of the big questions in cancer and how P-bodies can be the key to answering these questions.

R. *We thank the reviewer for appreciating the revised version. We have now mentioned how P-bodies could, at least partially, provide an answer to some of the unresolved questions in cancer research (lines 454-457).*

Introduction: Improved and more clearly demonstrating that condensates are operating at each level.

R. *Many thanks for the comment.*

P-bodies properties and biological functions: This is an important section for readers to connect with the rest of the sections. It is disjointed with a bunch of condensates that are not related or somewhat related to P-bodies. My suggestion is to split the section into smaller more purposeful sections:

1. Membrane-less organelles and phase separation: Short section laying out the concept underlying phase separation and membrane-less organelles including the no-P-body examples mentioned, as well as species specific condensates that are similar to P-bodies, but not P-bodies (GW bodies, P-granules, etc). You can end it with the P-body and stress granule verbiage in the first paragraph.
2. P-bodies background and historical context: Start with paragraph 2 of "P-bodies properties and biological functions" section, and gather any critical yeast information here. Note: In yeast, under homeostasis (log-phase growth), P-bodies aren't present, which is more similar to human stress granules. Human P-bodies are constitutively present. This section can finish with something similar to these two points separated in your current introduction: "In yeast, P-bodies assembly is primed by different environmental conditions, such as glucose deprivation, osmotic stress and DNA replication stress, but these events have not been confirmed in all eukaryotes. Since most of these studies have been made in yeast cells and the findings might not be extrapolated to mammalian systems, this review will primarily focus on the structure and regulation of mammalian P-bodies."
3. Key players of human P-bodies: Start with paragraph 3 of "P-bodies properties and biological functions" section and finish the section with human aspects of paragraphs 4 and 5. Maybe move the post-translational information into your "Control of P-body dynamics by post-translational modifications." section?

R. *We wish to thank the reviewer for the suggestions. We have now divided the paragraph in 4 sections:*

"Membrane-less organelles and liquid-liquid phase separation (LLPS)" (lines 68-95);

"P-bodies background and historical context" (lines 97-124);

"Key players of mammalian P-bodies" (lines 126-150);

"Regulation of P-body dynamics" (lines 152-175).

The information regarding yeast P-bodies have now been placed in "Background and early characterization of P-bodies" paragraph (lines 107-124) and those regarding post-translational mechanisms are in the section "Control of P-body dynamics by post-translational modifications".

To note that tyrosine 475 of EDC3 mentioned in the paragraph “Regulation of P-body dynamics” plays a role in the binding with other P-body proteins and should not be considered a phosphorylation acceptor site for post-translation modifications (lines 162-165)

Control of P-body dynamics by post-translational modifications: Useful section. Since this a general P-body PTM section, the next section can have cancer-related P-body PTM-containing sections like the “Regulation of P-body dynamics by signaling pathways in cancer” section.

R. *Many thanks for the comment.*

A role of P-bodies in cancer development and progression: Improved.

R. *Many thanks for the comment.*

Regulation of P-body dynamics by signaling pathways in cancer: No comments other than potentially positioning this section after the “Control of P-body dynamics by post-translational modifications” section above.

R. *We thank the reviewer for the comment; however, we believe it might be more appropriate to include a section providing an overview on the role of P-bodies in cancer (“A role of P-bodies in cancer development and progression”) before addressing the specific contribution of post-translational modifications of P-body components in cancer progression.*

P-bodies, hypoxia and metabolic reprogramming in cancer: Improved.

R. *Many thanks for the comment.*

A role of P-bodies in cancer stemness: Improved.

R. *Many thanks for the comment.*

P-body in cancer chemoresistance: Improved.

R. *Many thanks for the comment.*

Conclusions and perspectives: Improved.

R. *Many thanks for the comment.*

Reviewer #2 (Remarks to the Author):

Overall, the authors have made changes in line with the comments. They have included more seminal research citations and have provided citations involving P-bodies in the context of human cells to center the focus of the review more towards P-body regulation and function in mammalian cells. Moreover, they have clarified the relevance of information derived from experiments in yeast cells by providing studies in mammalian cells that support these results. Many sections have been reworded for clarity and now provide more evidence relating the interactions that take place within P-bodies, changes in P-body localized proteins, and alterations in P-body number to tumor growth, progression, and drug response. Additionally, specific information from sources has been clearly and concisely summarized, providing context for broader statements regarding the role of P-bodies in tumor progression and metastasis.

R. *We wish to thank the reviewer for appreciating the changes we have made*

Minor Comments:

1. Lines 38-59: The three levels of gene expression are described in how they

relate to LLPS and condensates-this provides helpful context. However, this section could benefit from a more concise description of the background of these three levels of gene expression and a clearer focus on how P-bodies are implicated in post-transcriptional control.

R. *We thank the reviewer for the comment. However, we believe that, in the Introduction, it is more appropriate to provide a general description of the three levels of regulation of gene expression, as previously suggested by the reviewers, and discuss the role of P-bodies in gene expression in the following sections.*

2. Lines 83-88: The authors describe how DDX6 in P-bodies plays a role in the negative regulation of P-body-stress granule docking. Then, mention other mRNP condensates that associate with the endoplasmic reticulum and nucleolus. It could be useful here to directly mention the association of U-bodies with P-bodies as detailed in source 28.

R. *We thank the reviewer for the suggestion. We have now mentioned the link between U-bodies and P-bodies (lines 79-82).*

3. Lines 191-196: The review cited here does not mention P-bodies specifically, but rather details the role of translational control in cancer cell plasticity. However, the statement made here is directly related to how P-bodies regulate the expression of oncogenes. A statement detailing the role of P-bodies in mRNA translation and protein synthesis alterations could be added here.

R. *Thanks for the comment. We have now included a statement clarifying the role of P-bodies in the regulation of mRNA translation (lines 198-200).*

4. The addition of Figure 1 provides a visual representation for the relationship between different types of condensates and their roles in the regulation of gene expression. To center this figure more on the role of P-bodies, more detail could be provided regarding different RBPs in P-bodies and how these contribute to post transcriptional regulation in the context of cancer.

R. *We thank the reviewer for the comment. Figure 1 is related to the Introduction section and describe more broadly the different levels of regulation of gene expression. We think that could be more appropriate to describe the components of P-bodies and their role in cancer in the figure 2 and 3.*

Reviewer #3 (Remarks to the Author):

The authors have responded to all the points raised and the article is acceptable as it stands.

R. *We thank the reviewer for the positive evaluation of our revised version.*